

# Rainfall erosivity estimation using gridded daily precipitation datasets

Maoqing Wang[1], Shuiqing Yin[1], Tianyu Yue[1], Bofu Yu[2], Wenting Wang[3,1]

[1]State Key Laboratory of Earth Surface Processes and Resource Ecology, Faculty of Geographical Science, Beijing Normal
University, Beijing, 100875, China
[2]Australian Rivers Institute, School of Engineering and Built Environment, Griffith University, Nathan, Qld 4111, Australia
[3]Zhuhai Branch of State Key Laboratory of Earth Surface Processes and Resource Ecology, Beijing Normal University at
Zhuhai, 519087, China

*Correspondence to*: Shuiqing Yin (Yinshuiqing@bnu.edu.cn)

**Abstract.** Rainfall erosivity is one of the most important factors incorporated into the empirical soil erosion models USLE
(Universal Soil Loss Equation) and RUSLE (Revised Universal Soil Loss Equation). Gridded precipitation datasets have been
widely used in the estimation of rainfall erosivity, whereas biases due to scale differences between gridded data and gauge
data have been ignored. Based on daily precipitation observations from over 2000 stations in China, as well as four widely
used gauge-based gridded daily precipitation datasets, CPC, GPCC, CN05.1 and NMIC, this study compared the probability
density functions (PDFs) of the gridded and gauge datasets using the skill score method, quantified the bias of rainfall erosivity
(including the R-factor and 1-in-10-year event rainfall erosivity) estimated using the gridded daily precipitation datasets from
that estimated using the gauge daily precipitation dataset based on the area reduction factor (ARF) method, and established
correction factors for rainfall erosivity maps generated from gridded datasets. The results showed that the gridded daily data
reduced the frequency of no-rain days and the intensity of heavy precipitation. In the eastern part of China, the grid-estimated
R-factor values were underestimated by 15−40 % compared with the gauge-estimated values, and the grid-estimated 1-in-10-
year event rainfall erosivity values were underestimated by 25−50 %, whereas in the western part of China, noticeable random
errors were introduced. The lower probability and intensity of the daily precipitation larger than the 90th percentile in the
gridded datasets were mainly responsible for the underestimation. CN05.1 was the most-recommended among the four datasets,
as it had the lowest mean relative error (MRE), and the accuracy was higher for the eastern part of China than for the western
part of China. The MREs were 16.1 % and 25.1 % for the R-factor after applying correction factors of 1.708 and 1.010,
respectively, for the eastern and western part of China. The 1-in-10-year event erosivity had larger correction factors and MREs
than did the R-factor, with the MREs being 22.1 % and 27.2 % after applying correction factors of 1.959 and 1.880, respectively,
for the eastern and western part of China. This study pointed out that in the applications of gridded precipitation datasets, the
empirical models established based on gauge precipitation data should not be used directly for the gridded data, or a bias
correction process needed to be considered for the model outputs.



## 1 Introduction

Precipitation is one of the most crucial components of the water cycle, and reliable precipitation datasets are the basis for characterizing the precipitation process and its impact on other earth surface processes (Kidd and Huffman, 2011). In general, there are two major types of datasets: gauge data and gridded data. Gauge observations provide relatively accurate
measurements of precipitation at site locations. However, they have obvious limitations due to poor spatial coverage in many parts of the world (New et al., 2001; Kidd and Huffman, 2011), and gauge data alone are often insufficient for many applications (Villarini and Krajewski, 2008; Kotlarski et al., 2017).

To improve the spatial and temporal coverage of precipitation observations, numerous gridded datasets with different spatiotemporal resolutions have been developed. These gridded datasets are based on the interpolation of gauge observations
using different spatial interpolation methods (Peterson and Vose, 1997; Hulme, 1992; Xie et al., 2007; Schamm et al., 2014), retrieval of satellite or weather radar data (Huffman et al., 2007; Joyce et al., 2004; Ashouri et al., 2015), merged gauge-satellite or gauge-radar precipitation products (Adler et al., 2003; Xie et al., 1997), and reanalysis data obtained by merging observations and numerical forecast products through data assimilation technology (Dee et al., 2011; Kobayashi et al., 2015; Kanamitsu et al., 2002). These gridded precipitation datasets can be applied in global and regional climatological studies
(Alexander et al., 2006; Sun et al., 2014; IPCC, 2014), climate model evaluations (Kotlarski et al., 2017; Prein and Gobiet, 2017), and, more relevant to this study, can be used as a vital meteorological input to various land surface process models, such as hydrological models, crop models, and soil erosion models (Yilmaz et al., 2005; Hong et al., 2007; Jiang et al., 2012; Vrieling et al., 2014). In addition to gridded precipitation products based on observations, gridded precipitation projections from global climate models (GCMs) and regional climate models (RCMs) have been widely used to drive land surface process
models for climate change impact assessments (Booij, 2005; Liu et al., 2014; Gosling and Arnell, 2016). This kind of research is of fundamental importance for our adaptation to climate change, as significant global warming has occurred since the industrial revolution and is projected to continue throughout this century (IPCC, 2014).

When applying gridded precipitation datasets, certain issues deserve special attention. Apart from interpolation errors and satellite retrieval bias, gridded data behave differently from gauge observations in probability distribution functions (PDFs) of
the daily precipitation amounts due to the intrinsic difference in spatial scales, which is common in gauge-based interpolation products, satellite products, reanalysis products and outputs of climate models (Xie et al., 2007; Chen et al., 2008; Shen et al., 2010; Tapiador et al., 2012). This is expected because the gridded value, which represents the average precipitation over a gridded cell (about $1-100 \text{ km}^2$), is supposed to differ from the gauge data, which represents the sampled value of a point ($100-1000 \text{ cm2}$). The differences in PDFs between gridded and gauge datasets need to be considered in the application of the datasets;
otherwise, they may lead to bias in model outputs (Zhang et al., 2011; Hofstra et al., 2012; Gehne et al., 2016).

One typical case is the use of gridded precipitation data in the estimation of rainfall erosivity. Rainfall erosivity represents the potential of rainfall and runoff to cause soil erosion, which is one of the factors incorporated in the widely used soil erosion model USLE and RUSLE (Wischmeier and Smith, 1978; Renard et al., 1997). The mean annual rainfall erosivity (the R-factor)



is defined as the mean annual sum of the event $EI_{30}$, which is calculated as the product of the total kinetic energy, E, and the

maximum 30-minute intensity, I30, of an event (Wischmeier and Smith, 1958). The calculation of $EI_{30}$ requires breakpoint precipitation data or data with short-term observation intervals and long timeseries, which are unavailable in many countries and regions. To solve this problem, many statistical models using annual, monthly or daily precipitation data have been developed (Yu and Rosewell, 1996; Zhang and Fu, 2003). It has been shown that models using daily precipitation data can be used to estimate three aspects of rainfall erosivity that are required in the ULSE and RULSE models: the R-factor, seasonal

distribution of rainfall erosivity and event $EI_{30}$ values with different return periods; therefore, daily models have been widely used in soil erosion assessments in China (Xie et al., 2016).

In practice, to generate a rainfall erosivity map that provides information for an entire area, including areas without gauge observations, two approaches can be used. The first is to calculate rainfall erosivity for individual stations and then spatially interpolate the values onto a predefined grid; the second is to generate gridded rainfall erosivity maps based on gridded

precipitation products directly (Yin et al., 2017). The former approach was chosen in most previous studies and has been proven to be highly accurate since the R-factor, as a long-term average annual value, has less spatial variability, so the spatial interpolation does not lead to large errors (Hong et al., 1997; Zhang et al., 2003). The second method has been used more frequently recently as gridded precipitation datasets are more readily available and easily used to update rainfall erosivity maps. For example, Lu and Yu (2002) generated a map of rainfall erosivity for Australia at a 0.05-degree resolution using a gauge-

based gridded daily precipitation dataset. Zhu and Yu (2015) obtained a map of rainfall erosivity at a 0.25-degree resolution for mainland China from a gauge-based gridded daily precipitation dataset produced by the National Meteorological Information Center (NMIC). In regions where gauge precipitation records are scarce, such as Africa, satellite precipitation products have become the preferred data source. For instance, Vrieling et al. (2010, 2014) analyzed the variation in rainfall erosivity across Africa using the Tropical Rainfall Measuring Mission (TRMM) satellite product with a 3-hour temporal

resolution and a 0.25-degree spatial resolution. In addition, since future changes in precipitation are likely to influence rainfall erosivity, the gridded precipitation outputs of GCMs and RCMs have been used to project future rainfall erosivity under climate change (Nearing, 2001; Zhang et al., 2010; Biasutti and Seager, 2015; Panagos et al., 2017; Borrelli et al., 2020).

However, due to the intrinsic differences between gridded data and gauge data, rainfall erosivity estimated from gridded data is likely to be biased if inappropriate methods are used. The largest problem is that the statistical relationship between daily or

monthly precipitation and rainfall erosivity, which is established from gauge data, cannot be used directly for gridded data. Nearing (2001) pointed out that when projecting future rainfall erosivity, the results from GCMs would not be compatible with the results from historical gauge records since an integrated average over the entire grid square could not be compared with a record from a specific location within the grid. Biasutti and Seager (2015) also explained that even though the grid of a downscaled dataset was quite fine, the coefficients of equations estimated from the gauge values would not necessarily be

appropriate for a gridded dataset. In many studies, this problem has been overlooked, and the resulting biases have not been well-quantified (Zhu and Yu, 2015; Teng et al., 2017). To make gridded results comparable with gauge results, statistical erosivity models should be recalibrated or even rebuilt on the grid scale (Biasutti and Seager, 2015), or for projections from



GCMs, the gridded precipitation outputs need to be spatially downscaled to points, from which the future rainfall erosivity values can be calculated and then interpolated to generate an R-factor map (Zhang et al., 2010; Shiono et al., 2013).

The major objectives of this study are to (1) quantify the biases of precipitation metrics and rainfall erosivity values estimated with four gauge-based gridded daily precipitation datasets (CPC, GPCC, CN05.1 and NMIC) based on gauge daily precipitation observations and (2) develop correction factors of the grid-estimated rainfall erosivity to minimize the bias. First, we compared four gridded datasets with gauge daily precipitation observations for 2243 meteorological stations in China in terms of the differences in the PDFs of the daily precipitation values, precipitation metrics and rainfall erosivity values

estimated from a power function daily model. Second, a set of correction factors was established by relating the rainfall erosivity values from the four gridded daily datasets to those from the state-of-the-art rainfall erosivity map for mainland China generated based on hourly data from 2396 stations (Yue et al., 2020b). These correction factors can be useful when applying gridded daily precipitation datasets in the estimation of rainfall erosivity and soil erosion in China.

## 2 Data and methods

### 2.1 Data

#### 2.1.1 Gauge-observed daily precipitation data

Daily precipitation data from 2481 meteorological stations, archived by the China Meteorological Administration (CMA), were used in this study. The longest overlapping period between the gauge data and four sets of gridded data was from 1 January 1982 to 31 December 2014. If the precipitation data were missing for more than six days in a month during the

overlapping period, the data from that station were excluded from this study (CMA, 2003). Finally, the gauge observations of 2243 stations from 1982 to 2014 were selected.

#### 2.1.2 Gridded daily precipitation data

In this study, four gridded daily datasets were used: two worldwide gridded daily precipitation datasets, the CPC (Climate Prediction Center) Unified Gauge-based Analysis of Global Daily Precipitation (hereinafter referred to as CPC; Xie et al.,

2007) and the GPCC (Global Precipitation Climatology Centre) First Guess Daily product (hereinafter referred to as GPCC; Schamm et al., 2014); and two nationwide datasets, CN05.1, developed by China Meteorological Administration (Wu et al., 2013), and the China Gridded Daily Precipitation Product, developed by the National Meteorological Information Center (hereinafter referred to as NMIC; Shen et al., 2010). Details about the four datasets are shown in Table 1.

**Table 1. Basic information on gridded daily precipitation datasets.**

| Data Source | CPC | GPCC | CN05.1 | NMIC |
|---|---|---|---|---|
| **References** | Xie et al., 2007 | Schamm et al., 2014 | Wu et al., 2013 | Shen et al., 2010 |





| Spatial resolution | $0.5° \times 0.5°$ | $1° \times 1°$ | $0.25° \times 0.25°$ | $0.5° \times 0.5°$ |
|---|---|---|---|---|
| Interpolation method | Optimal interpolation (OI) with anomalies | Ordinary block kriging with anomalies | Climatology—thin plate smoothing splines; anomaly—angular distance weighting | Optimal interpolation (OI) with anomalies |
| Coverage | Global land surface | Global land surface | China | China |
| Period | 1979.1.1−present | 1982.1.1−present | 1961.1.1−present | 1957.1.1−present |
| No. of stations | 1979−2005: more than 30000; 2006−present: more than 17000 | 6000~8000 in Global Telecommunication System (GTS) | More than 2400 | More than 2400 |
| No. of stations in China | more than 700 Meteorological sites across China and more than 1000 hydrological sites in the Yellow River Basin | More than 700 | More than 2400 | More than 2400 |

## 2.2 Method

### 2.2.1 Skill score

The skill score provided a quantitative measure of the similarity of the PDFs. The similarity of the PDFs of the daily precipitation amounts between gauge and gridded data was measured with the skill score ($S_{score}$) following the methods outlined in Perkins et al. (2007):

$$S_{score} = \sum_1^n \min(Z_g, Z_o) , \tag{1}$$

where $n$ is the number of bins used to calculate the PDF for a given station (the bin width was set to 0.5 mm, and $n$ was determined by bin width and maximum precipitation); $Z_g$ is the fraction of values in a given bin from the gridded data; and $Z_o$ is the fraction of values in the same bin from the gauge-observed data. $Z_g$ and $Z_o$ were calculated when the daily precipitation amount was $\geq$ 0.1 mm. This metric, being the sum of the minimum value of two frequency distributions, measured the common area between two PDFs. The $S_{score}$ ranged from 0 to 1; the closer its value was to 1, the more similar the two PDFs were.





### 2.2.2 Definition of the precipitation metrics, rainfall erosivity, and areal reduction factors

Four metrics were used to measure the performances of the different datasets for both the average precipitation conditions and the frequency or intensity of extreme events (Table 2). The metrics were recommended by the World Climate Research Programme/Climate Variability and Predictability (WCRP/CLIVAR) Expert Team on Climate Change Detection, Monitoring,
and Indices (ETCCDMI) (Alexander et al., 2006).

To estimate rainfall erosivity, we used a model of erosivity as a power function of the daily precipitation amount with a coefficient that varies seasonally as a sinusoidal function of the month (Xie et al, 2016):

$$R_{day,m} = 0.2686 \left[ 1 + 0.5412 \cos \left( \frac{\pi}{6} m - \frac{7\pi}{6} \right) \right] P_{day,m}^{1.7265}, \text{ when } R_{day,m} \geq 10mm, \tag{2}$$

where $m$ is the month, from 1 to 12; $R_{day,m}$ is the rainfall erosivity value for the day in the $m$th month; and $P_{day,m}$ is the daily
precipitation amount for the day in the $m$th month and was no less than 10 mm, which was the threshold of erosive daily precipitation. The daily rainfall erosivity values were accumulated for each month, and the R-factor was defined as the sum of the average monthly rainfall erosivity values (MJ·mm·hm$^{-2}$·h$^{-1}$·a$^{-1}$) over the study period.

In addition, the 1-in-10-year event EI$_{30}$, which was the event rainfall erosivity value with different return periods of 10 years (MJ·mm·hm$^{-2}$·h$^{-1}$), was also considered. Though from daily precipitation records, only daily rainfall erosivity could be
generated, Yin et al. (2019) found that there was a good linear relationship between event and daily rainfall erosivity values for corresponding return periods (the 1-in-10-year event EI$_{30}$ was approximately 1.17 times as much as the 1-in-10-year daily rainfall erosivity value). The generalized extreme value distribution (Smith, 2001) was first used to fit the annual series of the maximum daily rainfall erosivity values, and the 1-in-10-year daily rainfall erosivity value was then generated based on the calibrated extreme value distribution. Finally, the 1-in-10-year daily rainfall erosivity value was multiplied by the conversion
factor of 1.17 to obtain the 1-in-10-year event EI$_{30}$.

The precipitation metrics and rainfall erosivity values were first calculated for individual stations from gauge data and then interpolated into grids consistent with the four gridded datasets in terms of the spatial resolutions (Table 1), using ordinary kriging to obtain $METRICS_{sta}$ and $R_{sta}$. The metrics and rainfall erosivity values were calculated directly from the gridded data to obtain $METRICS_{gri}$ and $R_{gri}$. Used to measure the difference between the two methods, Area Reduction Factor (ARF)
was defined in this study as the ratio of $METRICS_{gri}$ over $METRICS_{sta}$ ($R_{gri}$ over $R_{sta}$) (Fowler et al., 2005; Chen and Knutson, 2008):

$$ARF = \frac{METRICS_{gri}}{METRICS_{sta}}, \text{ for precipitation metrics; } ARF = \frac{R_{gri}}{R_{sta}}, \text{ for rainfall erosivity}, \tag{3}$$

In all the spatial interpolation processes, leave-one-out cross validation was used to assess the accuracy of the interpolation method. PBIAS (percent bias), NSE (Nash-Sutcliffe coefficient of efficiency) and RMSE (root-mean-square error) were
calculated to examine the differences between the actual values calculated from gauge daily precipitation data and the values predicted by the kriging interpolation:





$$PBIAS = \frac{\sum_{i=1}^{N}(O_i - P_i)}{\sum_{i=1}^{N} O_i} \times 100 \% \,, \tag{4}$$

$$NSE = 1 - \frac{\sum_{i=1}^{N}(O_i - P_i)^2}{\sum_{i=1}^{N}(O_i - \bar{O})^2} \,, \tag{5}$$

$$RMSE = \sqrt{\frac{1}{N}\sum_{i=1}^{N}(O_i - P_i)^2} \,, \tag{6}$$

where $O_i$ is the rainfall erosivity value of the $i$th site calculated from the gauge-observed data and $P_i$ is the rainfall erosivity value of the $i$th site predicted by the kriging method from the cross-validation process. The closer NSE was to 1, and the closer PBIAS and RMSE were to 0, the better the interpolation model was. A negative PBIAS value indicated that the interpolation model overestimated the values at the sites, and conversely, a positive PBIAS value indicated that the model underestimated the values at the sites.

The ARFs in the eastern part of China, where the climate is relatively humid and soil erosion is mainly caused by water, may be different from those in the western part of China, where the climate is drier and soil erosion is mainly caused by wind and freeze-thaw processes. In addition, the different densities of meteorological stations could also influence the results. Therefore, the study area was divided into two parts (Chen and Zhu, 1989), and the evaluation was carried out in the two parts separately.

**Table 2. Definition of precipitation metrics.**

| Metrics | Description |
|---|---|
| **PRCPTOT** | Mean annual total precipitation from wet days |
| **WD** | Wet days: mean annual total days when precipitation ≥ 1 mm |
| **R95pTOT** | Mean annual total precipitation from days when precipitation > 95th percentile on wet days in the study period |
| **R×1 day** | Mean annual maximum 1-day precipitation amount |

**2.2.3 Correction factors for rainfall erosivity from gridded daily data**

Because of the lack of long-term precipitation data with high temporal resolution that can be used to calculate $EI_{30}$, the main method for obtaining nationwide rainfall erosivity datasets is based on daily precipitation data at present (Xie et al., 2016). To improve the accuracy of the rainfall erosivity map, Yue et al. (2020b) collected hourly precipitation data from more than 2000 stations and generated a state-of-the-art rainfall erosivity map for China. In this study, we considered the rainfall erosivity map

for the period of 1982−2014 generated from Yue's method as a reference to evaluate the accuracy of the rainfall erosivity values estimated from gridded daily precipitation data. MAE (mean absolute error) and MRE (mean relative error) were calculated for each grid box where a rain gauge was located:





$$MAE = \frac{1}{N}\sum_{i=1}^{N}\left|R_{gri}(i) - R_{ref}(i)\right|, \tag{7}$$

$$MRE = \frac{\sum_{i=1}^{N}\left|R_{gri}(i) - R_{ref}(i)\right|}{\sum_{i=1}^{N}R_{ref}(i)} \times 100\ \%, \tag{8}$$

where $R_{ref}$ is the rainfall erosivity extracted from the reference map. The map contained two aspects: the average annual rainfall erosivity (the R-factor) and the 1-in-10-year event $EI_{30}$. The spatial resolution of the rainfall erosivity map was converted to match those of the four gridded datasets through bilinear interpolation resampling.

To facilitate users in obtaining rainfall erosivity values as close to the accurate value as possible based on gridded daily data in China, we established correction factors applicable to the four gridded datasets using the reference map. Linear regression

analysis was conducted based on the characteristics of the errors. It was assumed that the grid boxes that contained meteorological sites that were used in the generation process of the gridded data had higher accuracy in the gridded daily precipitation datasets than grid boxes that did not contain meteorological sites; therefore, these grid boxes were selected for use in developing the correction factors. For the eastern part of China, 647 grid boxes with meteorological sites used in the generation process of the four gridded precipitation products were selected for the correction factors, and the remaining 1348

grid boxes with meteorological sites inside them were used for the evaluation of the correction factors. Similarly, for the western part of China, 167 grid boxes were selected for use developing the correction factors, and 81 grid boxes were used for the evaluation. Then, the linear relationships were established as:

$$R_{ref} = a \cdot R_{ref}, \tag{9}$$

where $a$ is the correction factor. The rainfall erosivity value of each grid box estimated from the gridded data was corrected by

applying $a$. Then, the MAE and MRE from Eq. (7) and Eq. (8) were calculated to assess the improved rainfall erosivity values once the corrections had been made.






- ● Stations used for the derivation of correction factors
- ● Stations used for the evaluation of correction factors
- ── Boundary between the eastern and western part of China

**Figure 1. Spatial distribution of the meteorological stations with daily precipitation observations and the division of the eastern and western part of China. All stations were used in the evaluation of differences between $METRICS_{sta}$ and $METRICS_{gri}$, as well as $METRICS_{sta}$ and $METRICS_{gri}$. The stations marked with red dots were used for the derivation of correction factors, and those with blue dots were used for the evaluation. The three marked stations (Beijing, Guangzhou and Yumen) were used as examples to compare the PDFs.**





## 3 Results

### 3.1 Comparison of PDFs between gridded and gauge data based on skill score

On the national scale, the skill score values of CN05.1 were the smallest among the four gridded datasets (2.5 % of stations with $S_{score} > 0.9$ and 42.2 % of stations with $S_{score} > 0.8$), whereas NMIC had the largest skill score values (75.2 % of stations with $S_{score} > 0.9$). From the perspective of spatial distribution, the skill scores in the western part of China, such as Xinjiang Province and Gansu Province, were relatively small, as well as those in the northern regions in the eastern part of China, such as the Loess Plateau, Hebei Province and Shandong Province. The skill scores in the southern regions in the eastern part of

China were relatively large.

**Figure 2. Spatial distributions of the skill scores for (a) CPC, (b) GPCC, (c) CN05.1, and (d) NMIC.**

Figure 3 shows the PDFs of the daily precipitation amounts of the gauge observations and four gridded datasets of Beijing,

Guangzhou and Yumen (Fig. 1), which represented conditions in the northern and southern regions in the eastern part of China





as well as the western part of China, respectively. In Beijing and Guangzhou, the four gridded datasets reduced no-rain days (0 mm) but increased drizzle precipitation (0–1 mm) and light precipitation (1–10 mm) compared to those of the original gauge observations. For erosive precipitation, the four gridded datasets demonstrated slight increases in the frequency of occurrence of precipitation in the 10–30 mm range and decreases in the frequency of occurrence of precipitation larger than
40 mm in Beijing and precipitation larger than 50 mm in Guangzhou. For Yumen, erosive precipitation events rarely occurred (PDF < 0.3 %), and no-rain days accounted for more than 90 % of the gauge observations; CPC, GPCC and CN05.1 decreased the frequency of occurrence of no-rain days and erosive days, whereas NMIC obtained an almost perfect estimation of no-rain and light precipitation days and increased the frequency of occurrence of erosive days. For these three places, the differences between CN05.1 and the gauge observations were considerable, especially for the no-rain and low-intensity precipitation
events in the northern and western regions, whereas NMIC showed similar PDFs to the gauge observations. Figure 3 also shows a comparison of the extreme daily precipitation amounts (90th, 95th, 99th percentiles and the maximum). As the percentile and extreme degree increased, the reduction in the daily precipitation amounts of CPC, GPCC and CN05.1 increased, whereas these percentiles based on NMIC were close to or even exceeded those of the gauge observations.









**Figure 3. Comparison of PDFs and the extreme daily precipitation amounts between the gauge observations and four gridded datasets: (a) PDFs, Beijing, (b) extreme daily precipitation amounts, Beijing, (c) PDFs, Guangzhou, (d) extreme daily precipitation amounts, Guangzhou, (e) PDFs, Yumen, and (f) extreme daily precipitation amounts, Yumen.**

### 3.2 Comparison of the precipitation metrics and rainfall erosivity values based on ARF

ARFs indicating the differences in the precipitation metrics obtained through the two approaches ($METRICS_{sta}$ and
$METRICS_{gri}$), as well as the rainfall erosivity ($R_{sta}$ and $R_{gri}$), are shown in Fig. 4. The cross-validation results showed that the accuracy of the spatial interpolation in the $METRICS_{sta}$ ($R_{sta}$) generation process was quite high (Table 3), which indicated that the differences between $METRICS_{sta}$ and $METRICS_{gri}$ ($R_{sta}$ and $R_{gri}$) were mainly produced during the upscaling of the gauge precipitation measurements to grids; that is, the discrepancy occurred because of the gridded precipitation products. In the eastern part of China, the gridded data generally conserved the average annual precipitation amounts well, but wet days
were overestimated by 10–30 %, which led to lower daily precipitation intensities than those seen in the gauge data. For erosive precipitation ($\geq$ 10 mm day$^{-1}$, not shown in Fig. 4), the total erosive precipitation amounts of CPC, GPCC, and CN05.1 were approximately 10–25 % lower than the amounts seen in the gauge data, whereas the erosive days were captured well, leading to a reduction in erosive precipitation intensity. For extreme precipitation metrics, compared with the R95pTOT, which comprehensively reflected the intensity and frequency of daily precipitation of the high percentile, the reduction in the average
annual maximum daily precipitation was more obvious. The medians of the ARFs for the R95pTOT of CPC, GPCC, CN05.1 and NMIC were 0.85, 0.93, 0.97 and 0.97, respectively, and those for R×1 day were 0.69, 0.78, 0.76 and 0.85, respectively. In terms of rainfall erosivity, the medians of the ARFs for the R-factors of CPC, GPCC, CN05.1 and NMIC were 0.63, 0.77, 0.75 and 0.85, respectively, and for the 1-in-10-year event rainfall erosivity (1-in-10-year EI$_{30}$), they were 0.49, 0.62, 0.55 and 0.71, respectively. The results revealed that rainfall erosivity was more affected by extreme precipitation intensity, and the 1-in-10-
year EI$_{30}$ amplified the differences between gridded and gauge data in extreme precipitation conditions (Table 4).

The ARFs in the western part of China generally had similar patterns to those in the eastern part of China but with greater spatial variability (Fig. 4). The medians of the ARFs for the R95pTOT of CPC, GPCC, CN05.1 and NMIC were 0.82, 0.88, 0.93 and 1.08, respectively, and those for R×1 day were 0.78, 0.81, 0.66 and 1.01, respectively. The characteristics of extreme precipitation resulted in a larger discrepancy between $R_{gri}$ and $R_{sta}$. The medians of the ARFs for the R-factors of CPC, GPCC,
CN05.1 and NMIC were 0.57, 0.67, 0.41 and 1.06, respectively, and those for the 1-in-10-year EI$_{30}$ were 0.54, 0.49, 0.31 and 0.70, respectively.

**Table 3. Cross-validation results for the precipitation metrics and rainfall erosivity values.**

| Dataset | PBIAS (%) | | | NSE | | | RMSE (the units were the same as those of the metrics) | | |
|---|---|---|---|---|---|---|---|---|---|
| | China | Eastern | Western | China | Eastern | Western | China | Eastern | Western |





| | | | | | | | | | |
|---|---|---|---|---|---|---|---|---|---|
| **PRCPTOT** | −0.10 | −0.11 | 0.31 | 0.99 | 0.96 | 0.85 | 95.7 | 95.6 | 96.4 |
| **WD** | 0.00 | −0.08 | 1.12 | 0.95 | 0.96 | 0.88 | 6.5 | 5.7 | 11.2 |
| **R95pTOT** | −0.07 | −0.08 | 0.26 | 0.99 | 0.95 | 0.80 | 30.4 | 31.2 | 23.7 |
| **R×1 day** | −0.01 | 0.03 | −1.01 | 0.99 | 0.92 | 0.76 | 8.46 | 8.74 | 5.59 |
| **R-factor** | −0.10 | −0.12 | 0.33 | 0.94 | 0.93 | 0.91 | 666.1 | 680.5 | 543.3 |
| **1-in-10-year $EI_{30}$** | 0.13 | 0.31 | −5.31 | 0.66 | 0.61 | 0.47 | 577.1 | 588.6 | 470.6 |











**Figure 4. ARFs for the precipitation metrics and rainfall erosivity values. The bars show the variation across the**
**stations, marking the median, Q1 and Q3 ranges (box), and the whiskers mark the range of Q1 − 1.5IQRs to Q3 +**
**1.5IQRs (dashes):(a) CPC in the eastern part of China, (b) CPC in the western part of China, (c) GPCC in the eastern**
**part of China, (d) GPCC in the western part of China, (e) CN05.1 in the eastern part of China, (f) CN05.1 in the western**
**part of China, (g) NMIC in the eastern part of China, and (h) NMIC in the western part of China.**

**Table 4. Overestimation (+) or underestimation (−) of precipitation metrics and rainfall erosivity values for the four**
**gridded datasets. The values were calculated by percent bias.**

| Dataset | CPC | | GPCC | | CN05.1 | | NMIC | |
|---|---|---|---|---|---|---|---|---|
| | Eastern | Western | Eastern | Western | Eastern | Western | Eastern | Western |
| **PRCPTOT** | −6.6 | −11.3 | 1.0 | 3.9 | 2.1 | 10.2 | 3.1 | 13.4 |
| **WD** | 25.9 | 11.7 | 20.5 | 14.2 | 35.1 | 57.4 | 16.7 | 12.1 |
| **R95pTOT** | −16.0 | −18.7 | −7.3 | −3.7 | −2.8 | −4.4 | −2.2 | 11.6 |
| **R×1 day** | −32.4 | −22.5 | −22.7 | −13.4 | −24.5 | −33.0 | −15.1 | 4.6 |
| **R-factor** | −37.1 | −40.2 | −23.0 | −14.2 | −25.7 | −47.7 | −13.7 | 12.5 |
| **1-in-10-year EI$_{30}$** | −50.0 | −63.8 | −34.4 | −64.2 | −42.0 | −78.5 | −25.5 | −48.3 |

**3.3 Correction factors for rainfall erosivity values estimated using gridded daily data**

Based on Yue's rainfall erosivity map, we conducted regression analysis for the eastern and western part of China
separately to establish correction factors for the R-factors and 1-in-10-year event EI$_{30}$ values using stations for
calibration. The linear equations and coefficients of determination, $R^2$, are shown in Fig. 5 and Fig. 6.

For the R-factor, in the eastern part of China, the results showed that the difference between the R-factor estimated
from the gridded daily data and that extracted from Yue's map was larger than the difference between the R-factors
estimated from the gridded daily data and that estimated from the gauge daily data; the daily erosivity model (Eq.
(2)) generally underestimated the R-factor by 10–20 % in the eastern part of China. The correction factors
(regression coefficients) of CPC, GPCC, CN05.1 and NMIC were 1.958, 1.570, 1.708, and 1.445, respectively. The
$R^2$ values for the four gridded datasets were all above 0.80, indicating that the linear models performed well in the
eastern part of China. However, in the western part of China, the estimated R-factors using CPC and CN05.1 were
almost unbiased, and the correction factors were close to 1, whereas for GPCC and NMIC, the correction factors





were 0.459 and 0.478, respectively. The $R^2$ values varied from 0.26 to 0.79, indicating that random errors played

more important roles in this region than those in the eastern part of China.

For the 1-in-10-year event $EI_{30}$ in the eastern part of China, the correction factors of CPC, GPCC, CN05.1 and NMIC were 2.133, 1.702, 1.959, and 1.477, respectively, showing that rainfall erosivity events caused by high-intensity and low-frequency precipitation events calculated using gridded data were more underestimated than those of the R-factor. In the western part of China, the estimated 1-in-10-year event $EI_{30}$ using CPC, GPCC and

NMIC were almost unbiased, whereas for CN05.1, the correction factor was 1.880.

To evaluate the performance of the correction factors, the MAEs and MREs of the stations used for evaluation were calculated before and after application of the correction factors. For the R-factor, in the eastern part of China, the MREs of CPC, GPCC, CN05.1, and NMIC were reduced by 28.4 %, 20.3 %, 23.2 % and 13.3 %, respectively, and CN05.1 had the smallest MRE (16.1 %, Table 5) after the correction. In the western part of China, the correction

factors of CPC and CN05.1 were close to 1, and the estimated R-factor of GPCC showed large random errors, so it was not necessary to conduct the correction except for on NMIC, whose MRE was reduced by 33.8 % with the correction. For the 1-in-10-year event $EI_{30}$, in the eastern part of China, the MREs of CPC, GPCC, CN05.1, and NMIC were reduced by 34.0 %, 20.1 %, 26.0 % and 17.1 %, respectively, and NMIC had the smallest MRE (19.4 %, Table 6) with the correction, followed by that of CN05.1 (22.1 %). In the western part of China, only CN05.1 was

worth correcting (the MRE was reduced by 21.0 %).

Taken together, the results showed that the correction factors of CN05.1 had the best performance on both the R-factor and 1-in-10-year event $EI_{30}$. Maps of rainfall erosivity in China were generated based on CN05.1 by applying the correction factors (Fig. 5 and Fig. 6). To solve the discontinuity near the boundary area between the eastern and western part of China caused by the difference in correction factors, a buffer was used for each region following

the method outlined in Yue et al. (2020a).







**Figure 5. Comparison of the R-factors estimated from the gridded data and those extracted from Yue's map. Due to the different spatial resolutions, the number of independent grids corresponding to stations used for the correction factor establishment in the four gridded datasets is different. (a) CPC in the eastern part of China using 417 grids, (b) CPC in the western part of China using 149 grids, (c) GPCC in the eastern part of China using 163 grids, (d) GPCC in the western part of China using 126 grids, (e) CN05.1 in the eastern part of China using 587 grids, (f) CN05.1 in the western part of China using 158 grids, (g) NMIC in the eastern part of China using 416 grids, and (h) NMIC in the western part of China using 149 grids.**







Figure 6. As in Fig. 5, but for the 1-in-10-year event $EI_{30}$.





**Table 5. Performance of the correction factors for the R-factor. In the eastern part of China, 805, 274, 1318, 805 validation grids were used for CPC, GPCC, CN05.1 and NMIC, respectively; in the western part of China, 77, 52, 84, 77 validation grids were used for CPC, GPCC, CN05.1 and NMIC, respectively.**

| | MAE (MJ mm hm$^{-2}$ h$^{-1}$ a$^{-1}$) | | | | MRE (%) | | | |
| Dataset | Eastern | | Western | | Eastern | | Western | |
| | Without corr. | With corr. | Without corr. | With corr. | Without corr. | With corr. | Without corr. | With corr. |
|---|---|---|---|---|---|---|---|---|
| CPC | 2297.1 | 954.3 | 113.3 | 130.0 | 48.6 | 20.2 | 29.7 | 34.1 |
| GPCC | 1792.7 | 846.2 | 112.6 | 209.4 | 38.5 | 18.2 | 30.9 | 57.4 |
| CN05.1 | 1862.3 | 762.4 | 98.3 | 97.1 | 39.3 | 16.1 | 25.4 | 25.1 |
| NMIC | 1510.1 | 880.1 | 289.3 | 160.3 | 31.9 | 18.6 | 75.8 | 42.0 |

**Table 6. As in Table 5, but for the 1-in-10-year event EI$_{30}$.**

| | MAE (MJ mm hm$^{-2}$ h$^{-1}$) | | | | MRE (%) | | | |
| Dataset | Eastern | | Western | | Eastern | | Western | |
| | Without corr. | With corr. | Without corr. | With corr. | Without corr. | With corr. | Without corr. | With corr. |
|---|---|---|---|---|---|---|---|---|
| CPC | 1227.4 | 488.2 | 90.1 | 83.6 | 56.4 | 22.4 | 33.3 | 30.9 |
| GPCC | 879.0 | 471.9 | 92.5 | 109.9 | 43.5 | 23.4 | 35.6 | 42.4 |
| CN05.1 | 1042.1 | 478.1 | 129.3 | 73.2 | 48.1 | 22.1 | 48.2 | 27.2 |
| NMIC | 794.9 | 423.2 | 70.2 | 83.3 | 36.5 | 19.4 | 26.0 | 30.8 |

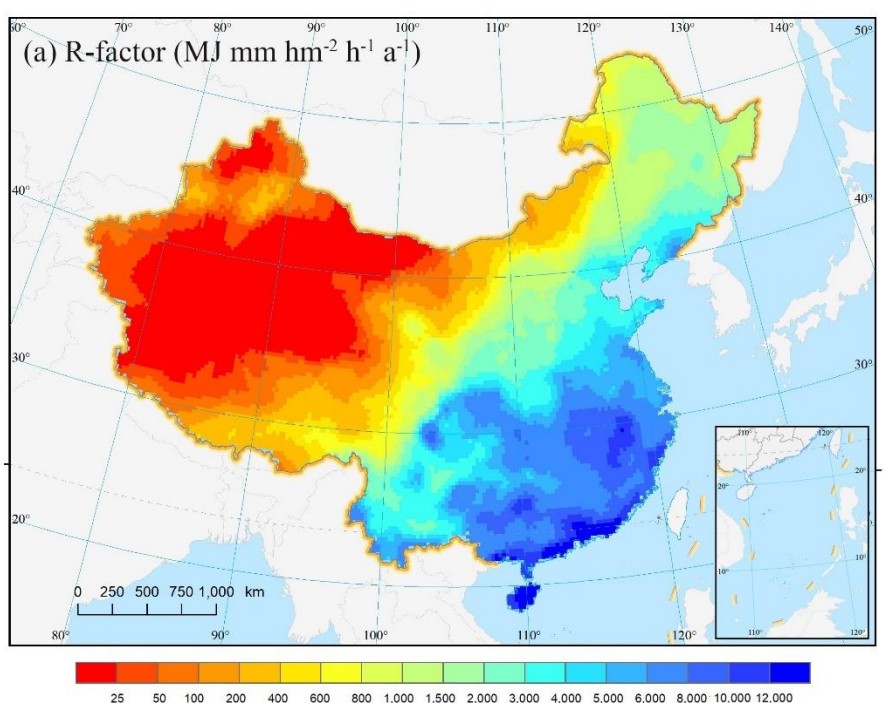

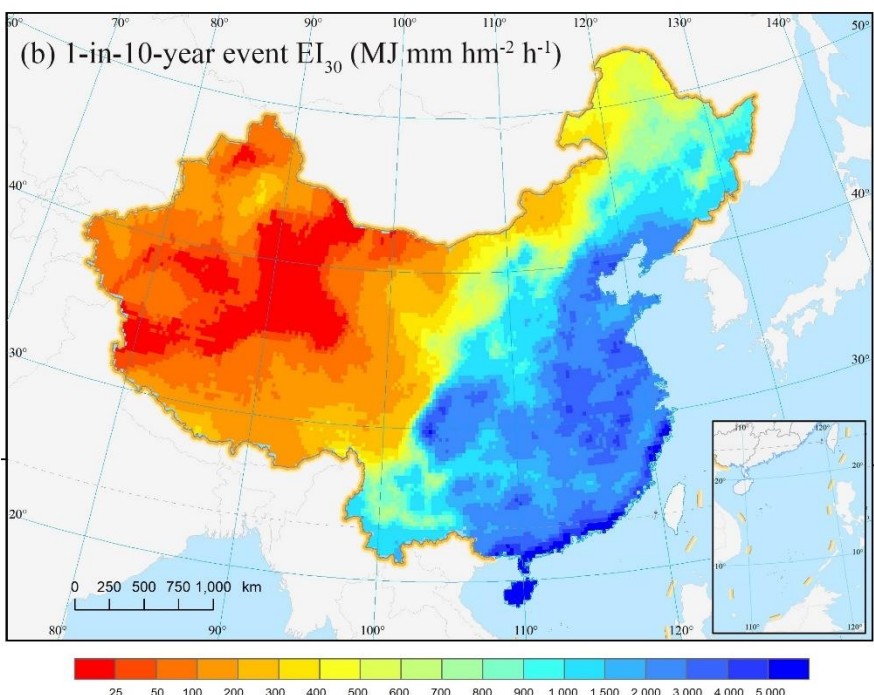

**Figure 7. Maps of the R-factor and 1-in-10-year event EI$_{30}$ values based on CN05.1 after applying the correction factors.**





## 4 Discussions

The gridded daily precipitation datasets have different PDFs of daily precipitation than does the gauge dataset. The main
difference is that the gridded datasets reduced the frequency of both no-rain days and heavy precipitation compared to those
of the original station observations. In the daily rainfall erosivity model, rainfall erosivity increased exponentially with the
increase in erosive daily precipitation amount, which meant that a reduction in heavy and extreme precipitation would result
in a more serious underestimation of rainfall erosivity from the gridded daily precipitation datasets, especially for the event
rainfall erosivity of high intensity and long return periods. The difference mainly comes from the spatial scale between the
grid and the point, the station density used for generating the gridded dataset, the interpolation method and the follow-up
bias correction method.

Although it is very difficult to define the ground truth for the PDFs of mean precipitation over the target grid boxes without a
very dense gauge network or reliable remote sensing data, it can be inferred that the spatial scale discrepancy between the
grid and point is objective and is influenced by precipitation characteristics and the spatial resolutions of the grids. The skill
score, which qualifies the similarity of PDFs between the grid and point scales, demonstrated larger values in the southern
regions in the eastern part of China. In western China and northern regions in eastern China, precipitation tends to be
concentrated in summer rainstorms, which are local and short-term, whereas in the southern regions in eastern China, there
tends to be continuous precipitation events covering a wide area and lasting a long time, such as the Meiyu, which results in
a smaller difference between the grid and point scales. It can be inferred that the differences between the grid and point
scales would decrease with decreasing grid size. One extreme case is that if the grid size is small enough to a point, a
difference would not exist.

The station density used to generate the gridded dataset, the interpolation method and the follow-up bias correction method
influenced the accuracy of the descriptions on the discrepancy between the spatial scales. Taking an extreme example, if the
station density is high enough and the areal precipitation in a grid is observed by all gauges in the grid, the true value of the
areal precipitation would be obtained, and the true difference of PDFs due to the spatial scale could be quantized. However,
this is not practical over a wide region, and the interpolation error was generated and increased with the decrease in station
density in the generation process of the gridded dataset. Among the four datasets in this study, CN05.1 and NMIC were
generated based on more than 2000 stations over mainland China, which used nearly three times the number of stations
compared with the other two datasets, CPC and GPCC. The increase in station density was believed to be of critical
importance in the interpolation of precipitation, especially for daily scales with high spatial variability. This finding can be
verified from the better performance of the gridded datasets in the eastern region, which has a higher station density (> 4
stations per 10,000 square kilometers) than does the western region (< 1 station per 10,000 square kilometers).

The largest difference between CN05.1 and NMIC was the different interpolation method, and NMIC applied a bias
correction process such as quantile mapping, which resulted in a much higher similarity of PDFs compared with other
datasets. Since the ground truth for the PDFs of the grid was hard to know, the scale difference of the PDFs between the grid





and point scale was difficult to quantify, it was doubtful whether the bias correction method adapted by NMIC was necessary and suitable. In addition, CN05.1 has the highest spatial resolution of $0.25° \times 0.25°$. To determine whether the spatial resolution affected the accuracy of the rainfall erosivity estimations, two spatial resolutions of $0.5° \times 0.5°$ and $1° \times 1°$ were derived from the CN05.1 dataset based on the bilinear interpolation method and were used to calculate rainfall erosivity. It

was found that the linear relationship between the results from the gridded data and gauge data tended to be better with the increasing spatial resolution; thus, the bias of rainfall erosivity estimated from CN05.1 resulting from the difference in PDFs of daily precipitation between the grid and the point scale could be better corrected. In summary, CN05.1 was the most-recommended dataset among the four gridded datasets for the estimation of rainfall erosivity in China.

Although rainfall erosivity maps based on hourly data are currently available (Yue et al., 2020b), they are not easy to update

in a timely manner since the collection of hourly data is harder than that of daily data. Gauge-based gridded daily precipitation datasets, such as CN05.1 in China, are easily available and can be used conveniently for the generation of rainfall erosivity maps (Zhu and Yu, 2015). Spatial scale discrepancies exist not only in gauge-based gridded precipitation data but also in satellite, merged and climate model-simulated precipitation products. This makes the direct use of gridded precipitation datasets dangerous in applications for which the empirical estimation method of a variable (rainfall erosivity, in

this study) is developed based on gauge-observed precipitation. If absolute values or absolute changes of the variable are calculated from the gridded dataset, it is suggested that the empirical model is rebuilt at the grid scale (Biasutti and Seager, 2015) or the systematic bias of the results generated from the model developed at the gauge scale are corrected. In this study, the latter method was adopted.

## 5 Conclusions

Based on the daily precipitation observations of over 2000 stations in China as well as four widely used gridded daily precipitation datasets, CPC, GPCC, CN05.1 and NMIC, this study compared the PDFs of the daily precipitation amounts, precipitation metrics and rainfall erosivity factors between the gridded daily datasets and the gauge daily dataset and established correction factors for the four gridded daily precipitation datasets using a high-precision rainfall erosivity map for China. The main conclusions are as follows:

(1) The PDFs of gridded daily data were different from that of the gauge data, mainly reflecting reductions in no-rain days and heavy precipitation days and increases in light precipitation days. NMIC had the most similar PDF with the gauge data (for 75.2 % of the stations, $S_{score} > 0.9$), whereas CN05.1 had the most different PDF (for 2.5 % of the stations, $S_{score} > 0.9$, and 42.2 % of the stations $> 0.8$).

(2) In the eastern part of China, the medians of the ARFs for the R-factors of CPC, GPCC, CN05.1 and NMIC were 0.63, 0.77,

0.75 and 0.85, respectively, and those for the 1-in-10-year $EI_{30}$ were 0.49, 0.62, 0.55 and 0.71, respectively. In the western part of China, the medians of the ARFs for the R-factors of CPC, GPCC, CN05.1 and NMIC were 0.57, 0.67, 0.41 and 1.06, respectively, and those for the 1-in-10-year $EI_{30}$ were 0.54, 0.49, 0.31 and 0.70, respectively. The results indicated that rainfall

erosivity values estimated using gridded daily data were significantly lower than those estimated using gauge daily data in most cases, which was mainly caused by reductions in extreme precipitation intensity in the gridded data, especially for the 1-

in-10-year $EI_{30}$.

(3) In the eastern part of China, correction factors for the R-factor of CPC, GPCC, CN05.1 and NMIC were 1.958, 1.570, 1.708, and 1.445, respectively, and those for the 1-in-10-year event $EI_{30}$ were 2.133, 1.702, 1.959, and 1.477, respectively. With the correction, for the R-factor, the MREs based on separate validation datasets were reduced by 28.4 %, 20.3 %, 23.2 % and 13.3 %, respectively, and for the 1-in-10-year event $EI_{30}$, the MREs were reduced by 34.0 %, 20.1 %, 26.0 % and 17.1 %,

respectively. There was no obvious improvement after applying the correction factors for the western part of China except the R-factor of NMIC and the 1-in-10-year event $EI_{30}$ of CN05.1. CN05.1 was the most-recommended dataset, with the lowest MREs of 16.1 % for the R-factor and 22.1 % for the 1-in-10-year event $EI_{30}$ in the eastern part of China and MREs of 25.1 % for the R-factor and 27.2 % for the 1-in-10-year event $EI_{30}$ in the western part of China after the application of correction factors.


*Data availability.* Gauge daily precipitation, CN05.1 and NMIC datasets were provided by the National Meteorological Information Center and the National Climate Center, CMA, for exclusive use in this study. CPC and GPCC datasets were download from the websites of the Climate Prediction Center [https://www.cpc.ncep.noaa.gov/products/monitoring_and_data/] and the Global Precipitation Climatology Centre [https://climatedataguide.ucar.edu/climate-data/gpcc-global-precipitation-

climatology-centre].

*Author contribution.* The study was designed and planned by MW, SY, and BY. MW carried out the analysis and wrote the manuscript. SY and BY guided the analysis methods and revised the draft. TY carried out the preliminary data analysis and helped the spatial interpolation process. WW downloaded the precipitation datasets and wrote the code for the calculation.


*Competing interests.* The authors declare that they have no conflict of interest.

*Acknowledgement.* This work was supported by National Key R&D Program (no.2018YFC0507006) and the National Natural Science Foundation of China (no. 41877068). We are grateful to the National Meteorological Information Center, and the

National Climate Center, CMA [https://www.ncc-cma.net/cn/], for providing us with CN05.1 and NMIC precipitation datasets. We also would like to thank the high-performance computing support from the Center for Geodata and Analysis, Faculty of Geographical Science, Beijing Normal University [https://gda.bnu.edu.cn/].



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
