# Peer review of "Rainfall erosivity estimation using gridded daily precipitation datasets"

_Hydrology and Earth System Sciences, 2020_

## Referee Comment (RC1) · Anonymous Referee #1 · 9 Jan 2021

This paper is well written and has an interesting objective. As the authors state, gridded precipitation products are increasingly used in environmental applications but may have significant biases because of spatial averaging. My primary criticisms have to do with methodology and the approach taken, though the analysis itself is well done.

Line 143 (equation 2): Some discussion of how this equation relates to RUSLE/RUSLE2 definitions for rainfall erosivity and criteria for erosive events might be helpful for making comparisons.

Line 156: I do not see a need for interpolating the gauge data to coincide with the grid locations. Presumably, none of the gauge locations happened to coincide with the grid locations, so basically, all observed/reference data came from interpolation, which introduces interpolation error. The gridded products actually represent cells (which the

authors nicely explain on line 57), and every gauge location is inside one of these cells such that the gauge values can be paired with the cell values. I suppose in eastern China with high data density, interpolation error isn't a problem, but in western China, it seems like it would be more of an issue. Is there a reason for doing it this way that could be clarified? How far away, on average, are the gauge stations from the grid points?

Line 190-193: Why resample the Yue et al. (2020b) map to the spatial resolutions of the gridded products? Doing this means that the correction factors are based on a comparison of a spatially averaged erosivity map to spatially averaged gridded climate data. So, it seems applying the determined bias correction factor to the gridded products doesn't eliminate the effects of spatial averaging, which I got the impression was an objective of the study. It seems to me that the Yue et al. (2020b) map should not be resampled; rather, the map should be sampled at its original resolution at the grid point locations.

Line 203 (equation 9): Rref is used twice (typo). In my opinion, it makes more sense for the equation to be Rgri=a·Rref so that the observed/reference data would be on the x-axis (the opposite is done in this paper). Linear regression assumes error is distributed along the y-axis (which should be the axis with gridded values), and in the calculation of slope, the variance of the x-axis data (which should be the reference values) standardizes the covariance of x and y. If the regression is done this way, the bias correction becomes the reciprocal of the slope.

Line 215-216: Normally there isn't a space between a percentage and the percentage sign.

---

## Referee Comment (RC2) · Anonymous Referee #2 · 11 Jan 2021

In the submitted paper, authors investigated relationship between gauge data and gridded daily precipitation datasets. Multiple variables were used for the comparison with focus on the precipitation and rainfall erosivity. The topic of the paper could potentially be of interest for readers of this journal. However, there are several drawbacks related to the submitted manuscript that should be improved before further evaluation of this manuscript.

Most importantly, based on the presented results (Table 5) and second aim of the study (i.e. develop a correction factors) I think that authors should perform additional investigations in order to fulfil this second aim since according to Table 5, the developed correction factors do not lead to improved results (at least not for all cases).

Additionally, there are multiple parts that should be either better explained or enhanced

(some specific comments are listed below). More specifically, I am missing a take home message that could be useful for the international readership. Authors state that correction factors need to be applied but the factors they developed have local characteristics and do not even improve the results in all investigated cases.

Moreover, it should be noted that there are already quite some things done in relation to the rainfall erosivity assessment, even at global scale (for example Global rainfall erosivity assessment based on high-temporal resolution rainfall records by Panagos et al., 2017) using high-temporal resolution data. Thus, this assessment of the erosivity using daily data (either gridded or point-observed) should be well justified. Also in China you have a nice network of high-temporal resolution data. Additionally, there are also some satellite products already developed that have sub-daily temporal resolution. Thus, I am missing a better justification of using of daily data because also for example Yue et al. 2020b map is based on the hourly data. Thus, why would one need to estimate the erosivity based on daily data if a map based on hourly data is already developed and available? Why dealing with daily data since such estimates of erosivity (based on daily data) should only be used in cases without hourly or sub-hourly data because they are less accurate.

Some specific comments:

L76: how can a spatial map be highly accurate since no information about the Âżactu- alÂń R-factor is available. In order to obtain a value that is as close to actual drop-size-distribution measurements are needed, which can only performed for specific station.

L116: I think that more detailed description of the gauge data should be provided. What is the equipment used, is the data verified, what is the data quality, anything that have an effect on the results of this study should be included.

Table 1: What is the number of stations in the period 2006-present for the CPC, more than 17000 or more than 700?

Eq. 2: Why a threshold of 10 mm is used if standard RUSLE threshold is 12.7 mm or 6.25 mm in 15 min?

L158-159: These variables should be better explained and these sentences should be rewritten.

Eq. 3: Can ARF be defined twice using different variables?

Table 2: What is the difference between mean annual precipitation from only wet days or from both wet and dry days?

Eq. 9: I do not understand this equation, Rref is used on left and right side? Thus, a can only be 1?

Figure 2: The readability of these figures is too low.

Figure 3: Can you really say that these are PDFs? You are showing number of rainfall events in different bins? Or at least better visual presentation should be made since it is not easy to see which dataset yields better agreement with observed data. Additionally, can you add a summary of these differences between models and observed data? Thus, which model/dataset yields the best fit to the observed data.

Figure 4: I am sorry but I cannot understand this figure since obviously I do not understand correctly what should the ARF be according to your study. Perhaps this is related to the definition in L158-162 and Eq. 3 that should be improved. What is usually defined as areal reduction factor can be seen here (for example): https://www.sciencedirect.com/science/article/abs/pii/S0022169418301999. Thus, something different that is shown in Figure 4. For example, the ARF should be a value between 0 and 1. I suggest that authors try to make this a bit easier to understand (what is shown here) for the reader.

L289: Any specific reason for such behavior?

Figure 5: The resolution of this figure is quite low. I would suggest to add the number of

points in all plots, since it seems that some cases (h) or (d) have relatively low number of points compared to the size of the investigated area. Or is this already written in the caption where "grids" is used? Moreover, grids or grid cells?

Table 5: Correction in some cases leads to worse results? What is then the rationale behind adopting such "corrections" factors if the final result is even worse than without these factors.

Figure 7: I suggest to add a map that shows the difference between the erosivity map after applying correction and the Yue map.

Discussion: What not merging results and discussion since you already have some discussion in the results section? And then perhaps also the results section would be more easier to read and understand.

L385: "Reductions": gauge data compared to grid data or grid data compared to gauge data?

L400: What is the purpose of using a correction factor if it does not yield improved performance? Is there any alternative, a better method that should be elaborated?

Conclusions: What are the practical conclusions of this study that could be useful for people dealing with rainfall erosivity in other parts of the world? What is the main take home message?

---

## Referee Comment (RC3) · Anonymous Referee #3 · 12 Jan 2021

This paper is well written with appropriate methods and datasets. However, the focus is on the comparison of different gridded rainfall datasets rather than the actual estimation (methods) on rainfall erosivity. Thus, I suggest to narrow down the title to truly reflect the actual content of the manuscript, the current title (Rainfall erosivity estimation using gridded daily precipitation datasets) is too broad. I also suggest the authors to clearly specify the differences of this research from previous studies and the implications to international readers, and add the aims and objectives in the last paragraph under the Introduction section. Some minor issues for authors consideration: Line 203: please differentiate the parameters in Eq. 9 ($R_{ref} = a\ R_{ref}$) Fig. 3-6: improve/change the fonts to make the figures clearer Check the unit for the R-factor: is hm-2 really km-2 or just use ha.

---

## Author Comment (AC1) · 5 Mar 2021

Reviewer's original comment in black
Our response to each comment in blue

This paper is well written and has an interesting objective. As the authors state, gridded precipitation products are increasingly used in environmental applications but may have significant biases because of spatial averaging. My primary criticisms have to do with methodology and the approach taken, though the analysis itself is well done.

Thank you for the review and most valuable comments and suggestions. We have carefully considered each of the comments and outlined, following each comment, how we will address these comments and how the manuscript will be revised.

Line 143 (equation 2): Some discussion of how this equation relates to RUSLE/RUSLE2 definitions for rainfall erosivity and criteria for erosive events might be helpful for making comparisons.

As mentioned in Line 65-70, the equation that relates daily precipitation amounts to rainfall erosivity was developed because of a lack of high-temporal resolution, typically 5-15min, precipitation data. Xie et al. (2016, Line 535 in the manuscript) evaluated the model and showed that the equation is accurate enough to estimate the mean annual rainfall erosivity and its seasonal variations when the erosive daily rainfall threshold was set to 10 mm. We will explain this point further in the revised manuscript.

Line 156: I do not see a need for interpolating the gauge data to coincide with the grid locations. Presumably, none of the gauge locations happened to coincide with the grid locations, so basically, all observed/reference data came from interpolation, which introduces interpolation error. The gridded products actually represent cells (which the authors nicely explain on line 57), and every gauge location is inside one of these cells such that the gauge values can be paired with the cell values. I suppose in eastern China with high data density, interpolation error isn't a problem, but in western China, it seems like it would be more of an issue. Is there a reason for doing it this way that could be clarified? How far away, on average, are the gauge stations from the grid points?

Precipitation metrics and rainfall erosivity values for individual stations were interpolated using ordinary kriging method, and then resampled to the corresponding spatial resolutions of the gridded precipitation products. Therefore, the comparison was done between the two kinds of average grid cell values. The leave-one-out crossvalidation results show that the accuracy of the spatial interpolation process is quite high and the interpolation errors are acceptably small since the precipitation metrics and rainfall erosivity values have less spatial variability (Table 3), though as pointed out, in the western China the interpolation errors are larger. Besides, the resampling process to obtain areal average values also did not introduce large errors.

We will revise the objectives of this study in the revised version to make these clearer as we have found that the objectives were not clearly stated in the manuscript as follows: (1) to contrast the gridded daily precipitation products with gauge data in terms of PDFs and extreme precipitation amounts, and to evaluate the smoothing effect of interpolation when areal precipitation for grid cells were generated using point (gauge) observations;
(2) to evaluate the magnitude of underestimation of rainfall erosivity calculated using gridded daily precipitation products compared with that produced by spatial interpolation of rainfall erosivity computed using point (gauge) observations;
(3) to establish bias correction factors to improve the accuracy of rainfall erosivity maps where only gridded precipitation products were available for estimating rainfall erosivity over large areas.

The interpolation and resampling process was done because one of our objectives is to compare the two approaches to obtain rainfall erosivity maps: (1) by calculating rainfall erosivity using gauge observations, and then interpolate the gauge-based erosivity values into different spatial resolutions (spatially averaged rainfall erosivity); (2) by calculating rainfall erosivity from gridded precipitation products directly. Therefore, the comparison needs to be made at a commensurate spatial scale. The same applied to other precipitation metrics. We will make it clearer in the revised version.

Line 190-193: Why resample the Yue et al. (2020b) map to the spatial resolutions of the gridded products? Doing this means that the correction factors are based on a comparison of a spatially averaged erosivity map to spatially averaged gridded climate data. So, it seems applying the determined bias correction factor to the gridded products doesn't eliminate the effects of spatial averaging, which I got the impression was an objective of the study. It seems to me that the Yue et al. (2020b) map should not be resampled; rather, the map should be sampled at its original resolution at the grid point locations.

As explained above, one objective of this study is to compare the two approaches to

obtain rainfall erosivity maps and establishing bias correction factors that can be used in China, not to eliminate the effects of spatial averaging.

Line 203 (equation 9): Rref is used twice (typo). In my opinion, it makes more sense for the equation to be Rgri=a_Rref so that the observed/reference data would be on the x-axis (the opposite is done in this paper). Linear regression assumes error is distributed along the y-axis (which should be the axis with gridded values), and in the calculation of slope, the variance of the x-axis data (which should be the reference values) standardizes the covariance of x and y. If the regression is done this way, the bias correction becomes the reciprocal of the slope.

There was a typo, and the equation 9 should have been $R_{gri} = a \cdot R_{ref}$. Thank you for pointing this out.

It is true that in general the x-axis is used for observed/reference values. Here our objective is to use gridded products data and equation (2) to estimate a biased R-factor first. The bias can then be largely removed by multiplying an adjustment factor. This adjustment factor happens to the slope of the regression line if we plot the reference R-factor on the y-axis.

Plot original method to do the regressions for the following reason:
The objective of this study is when people have gridded daily precipitation data, they can multiply the R value so computed with the slope to obtain the equivalent R as the values from the reference map. Plotting $R_{ref}$ on the y-axis can establish the linear regression model through the origin and obtain the slope using ordinary least squares regression (OLS):

$$Slope = \frac{\sum_{i=1}^{N} x_i y_i}{\sum_{i=1}^{N} x_i^2}$$

the slope of the regression is an unbiased estimate of the correction factor. If we flip the axis, the bias correction becomes the reciprocal of the slope but may not remain unbiased.

Line 215-216: Normally there isn't a space between a percentage and the percentage sign.
A good point, and we will revise the manuscript accordingly.

---

## Author Comment (AC2) · 5 Mar 2021

Please see attached .pdf supplement for response to RC2. Reviewer comments have been copied in black font. Our responses are written in blue font. For the figures that are not clearly displayed in the manuscript, we have attached high-resolution images. Figure 7 and the captions of Figure 3, 4, 5 and 6 have been revised according to the comments.

Please also note the supplement to this comment:
https://hess.copernicus.org/preprints/hess-2020-633/hess-2020-633-AC2-supplement.pdf

[Figure]

Figure 2. Spatial distributions of the skill scores for (a) CPC, (b) GPCC, (c) CN05.1, and (d) NMIC.

**Fig. 1.**
[Figure]

Figure 3. Comparison of the frequency distribution histograms of daily precipitation amounts and the extreme daily precipitation amounts between the gauge observations and four gridded datasets: (a) histogram, Beijing, (b) extreme daily precipitation amounts, Beijing, (c) histogram, Guangzhou, (d) extreme daily precipitation amounts, Guangzhou, (e) histogram, Yumen, and (f) extreme daily precipitation amounts, Yumen.

**Fig. 2.**

[Figure]

**Figure 4.** *Ratios* for the precipitation metrics and rainfall erosivity values. The bars show the variation across the stations, marking the median, Q1 and Q3 ranges (box), and the whiskers mark the range of Q1 − 1.5IQRs to Q3 + 1.5IQRs (dashes):
(a) CPC in the eastern part of China, (b) CPC in the western part of China, (c) GPCC in the eastern part of China, (d) GPCC in the western part of China, (e) CN05.1 in the eastern part of China, (f) CN05.1 in the western part of China, (g) NMIC in the eastern part of China, and (h) NMIC in the western part of China.

**Fig. 3.**

[Figure]

**Figure 5. Comparison of the R-factors estimated from the gridded data and those extracted from Yue's map. Due to the different spatial resolutions, the number of independent grid cells corresponding to stations used for the correction factor establishment in the four gridded datasets is different.**

(a) CPC in the eastern part of China using 417 grid cells, (b) CPC in the western part of China using 149 grid cells, (c) GPCC in the eastern part of China using 163 grid cells, (d) GPCC in the western part of China using 126 grid cells, (e) CN05.1 in the eastern part of China using 587 grid cells, (f) CN05.1 in the western part of China using 158 grid cells, (g) NMIC in the eastern part of China using 416 grid cells, and (h) NMIC in the western part of China using 149 grid cells.

**Fig. 4.**

[Figure]

[Figure]

Figure 6. Comparison of the 1-in-10-year event EI$_{30}$ estimated from the gridded data and those extracted from Yue's map. Due to the different spatial resolutions, the number of independent grid cells corresponding to stations used for the correction factor establishment in the four gridded datasets is different.

(a) CPC in the eastern part of China using 417 grid cells, (b) CPC in the western part of China using 149 grid cells, (c) GPCC in the eastern part of China using 163 grid cells, (d) GPCC in the western part of China using 126 grid cells, (e) CN05.1 in the eastern part of China using 587 grid cells, (f) CN05.1 in the western part of China using 158 grid cells, (g) NMIC in the eastern part of China using 416 grid cells, and (h) NMIC in the western part of China using 149 grid cells.

**Fig. 5.**

[Figure]

**Figure 7.** Spatial distribution of the R-factor and 1-in-10-year event EI30 using CN05.1 with bias correction factors (a and c), and the difference (b and d) in comparison with the original R-factor map (Yue et al., 2020b).

**Fig. 6.**

**Supplement:**

Reviewer's original comment in black
Our response to each comment in blue

In the submitted paper, authors investigated relationship between gauge data and gridded daily precipitation datasets. Multiple variables were used for the comparison with focus on the precipitation and rainfall erosivity. The topic of the paper could potentially be of interest for readers of this journal. However, there are several drawbacks related to the submitted manuscript that should be improved before further evaluation of this manuscript.

Thank you for the review and most valuable comments and suggestions. We have carefully considered each of the comments and outlined, following each comment, how we will address these comments and how the manuscript will be revised.

Most importantly, based on the presented results (Table 5) and second aim of the study (i.e. develop a correction factors) I think that authors should perform additional investigations in order to fulfil this second aim since according to Table 5, the developed correction factors do not lead to improved results (at least not for all cases).

Additionally, there are multiple parts that should be either better explained or enhanced (some specific comments are listed below). More specifically, I am missing a take home message that could be useful for the international readership. Authors state that correction factors need to be applied but the factors they developed have local characteristics and do not even improve the results in all investigated cases.

In the revised manuscript, we will highlight the inherent underestimation of rainfall erosivity when gridded precipitation products are used to compute this, and provide the bias correction factors for areas where gridded precipitation products remain the only data source. For regions where point (gauge) data are available to compute rainfall erosivity as in the case of China, bias correction factors are not needed.

We will revise the objectives of this study in the revised version to make these clearer as we have found that the objectives were not clearly stated in the manuscript as follows:

(1) to contrast the gridded daily precipitation products with gauge data in terms of PDFs and extreme precipitation amounts, and to evaluate the smoothing effect of interpolation when areal precipitation for grid cells were generated using point (gauge) observations;

(2) to evaluate the magnitude of underestimation of rainfall erosivity calculated using

gridded daily precipitation products compared with that produced by spatial interpolation of rainfall erosivity computed using point (gauge) observations;

(3) to establish bias correction factors to improve the accuracy of rainfall erosivity maps where only gridded precipitation products were available for estimating rainfall erosivity over large areas.

We will also highlight the take home messages in the Discussions and Conclusions sections as follows:

For rainfall erosivity estimation in other parts of the world:

(1) since gridded daily precipitation products are average values over grid cells, they are inherently less extreme compared with gauge observations, the R-factor calculated from gridded products are systematically underestimated by 10-40% compared with that generated by spatial interpolation of the gauge calculated values, and the magnitude of underestimation is larger (30-50%) for the 1-in-10-year event $EI_{30}$;

(2) it is better to use gauge observations to estimate rainfall erosivity where gauge data are available. When only gridded daily precipitation products are available, a bias correction factor ranging from 1.4~2.0 could be used to improve the accuracy of estimated rainfall erosivity. Without bias correction, direct application of gridded precipitation products would lead to systematic underestimation of rainfall erosivity in most parts of the world.

For wider research community:

(1) because the gridded precipitation products (represent areal average precipitation over a grid cell) are generally unable to capture the extreme precipitation amount, compared to gauge observations, care needs to be taken to avoid systematic bias when using gauge-based, satellite-based gridded precipitation products, reanalysis products and outputs of climate models as inputs to nonlinear models often found in hydrology and agriculture;

(2) when empirical models, such as the daily rainfall erosivity model considered in this study, are developed based on point observations, and gridded precipitation is the only available data source, bias correction is absolutely needed, or alternatively the model needs to be re-calibrated at a commensurate spatial scale.

Moreover, it should be noted that there are already quite some things done in relation to the rainfall erosivity assessment, even at global scale (for example Global rainfall erosivity assessment based on high-temporal resolution rainfall records by Panagos et al., 2017) using high-temporal resolution data. Thus, this assessment of the erosivity

using daily data (either gridded or point-observed) should be well justified. Also in China you have a nice network of high-temporal resolution data. Additionally, there are also some satellite products already developed that have sub-daily temporal resolution. Thus, I am missing a better justification of using of daily data because also for example Yue et al. 2020b map is based on the hourly data. Thus, why would one need to estimate the erosivity based on daily data if a map based on hourly data is already developed and available? Why dealing with daily data since such estimates of erosivity (based on daily data) should only be used in cases without hourly or sub-hourly data because they are less accurate.

As mentioned in Line 369 in the discussion section, though rainfall erosivity maps based on high-temporal resolution rainfall records are currently available, they are not easy to update in a timely manner since the collection of high-temporal resolution rainfall data is harder. In China the sub-daily precipitation observation datasets are not freely shared with ordinary users. The current rainfall erosivity assessments based on high-temporal resolution rainfall records generally use fewer stations in China, for example, high resolution rainfall records from only 387 stations in China were used in the study of Panagos et al., 2017. Many satellite products or radar rainfall products have sub-daily temporal resolution, but these products tend to have large bias and short time series. In many regions of the world, daily data are still the best choice considering both availability and accuracy. Besides, when predicting the future rainfall erosivity, the daily precipitation outputs from the climate model are much more reliable than the hourly or sub-hourly precipitation outputs.

Some specific comments:
L76: how can a spatial map be highly accurate since no information about the actual R-factor is available. In order to obtain a value that is as close to actual drop-size distribution measurements are needed, which can only performed for specific station.

Accurate R-factor values depend the rainfall intensity-kinetic energy relationship that is site-specific. In the manuscript, 'highly accurate R-factor' was meant to indicate that direct interpolation of gauge-based R-factor values would be more accurate than that using gridded precipitation products generated through interpolation of point rainfall observations. We will revise these sentences when revising the manuscript.

L116: I think that more detailed description of the gauge data should be provided. What is the equipment used, is the data verified, what is the data quality, anything that have

an effect on the results of this study should be included.

Thanks for your advice. We will add more details of the gauge data in the revised version.

Table 1: What is the number of stations in the period 2006-present for the CPC, more than 17000 or more than 700?

Sorry for our mistake. After reviewing the references, we have corrected Table 1 as follows:

**Table 1. Basic information on gridded daily precipitation datasets.**

| Data Source | CPC | GPCC | CN05.1 | NMIC |
|---|---|---|---|---|
| References | Chen et al., 2008 | Schamm et al., 2014 | Wu et al., 2013 | Shen et al., 2010 |
| Spatial resolution | 0.5° × 0.5° | 1° × 1° | 0.25° × 0.25° | 0.5° × 0.5° |
| Interpolation method | Optimal interpolation (OI) with anomalies | Ordinary block kriging with anomalies | Climatology—thin plate smoothing splines; anomaly—angular distance weighting | Optimal interpolation (OI) with anomalies |
| Coverage | Global land surface | Global land surface | China | China |
| Period | 1979.1.1–present | 1982.1.1–present | 1961.1.1–present | 1957.1.1–present |
| No. of stations | 1979–2005: ~30000; 2006–present: ~17000 | 6000~8000 in Global Telecommunication System (GTS) | ~2400 | ~2400 |
| No. of stations in China | ~200 | ~200 | ~2400 | ~2400 |

Eq. 2: Why a threshold of 10 mm is used if standard RUSLE threshold is 12.7 mm or 6.25 mm in 15 min?

Xie et al. (2016, Line 535 in the manuscript) evaluated the accuracy of the daily model (Equation 2) and reported that when the erosive daily rainfall threshold was set to 12 mm, average annual erosivity values were 1.1–6.2% lower than measured values defined in the USLE. When the erosive daily rainfall was set to be 9.7 mm, the average relative deviation reached the minimum (–0.1%). When the daily erosive rainfall threshold was set to be 10 mm, the average relative deviation was –0.5%, which has a

negligible difference from that when the threshold was set to be 9.7 mm. Besides, the threshold of 10mm corresponds to the demarcation between light and moderate rain stipulated by the China Meteorological Administration (CMA, 2003). For ease of use, the erosive daily rainfall threshold was determined to be 10 mm.

L158-159: These variables should be better explained and these sentences should be rewritten.
Eq. 3: Can ARF be defined twice using different variables?

Thanks for your advice. We will make these sentences clearer and change the index name, considering your comments on Figure 4.

Table 2: What is the difference between mean annual precipitation from only wet days or from both wet and dry days?

*Wet days* here is defined as the days when precipitation $\geq$ 1 mm to correspond to *WD*. We will revise the definition of *PRCPTOT*: Mean annual total precipitation from days when precipitation $\geq$ 1 mm.

Eq. 9: I do not understand this equation, Rref is used on left and right side? Thus, a can only be 1?

Sorry for our mistake. The equation 9 should be $R_{gri} = a \cdot R_{ref}$.
This typo will be corrected in the revised manuscript.

Figure 2: The readability of these figures is too low.

For the figures that are not clearly displayed in this manuscript, we will attach high-resolution images to the comments. If they are still hard to read and understand, we will consider changing the visual presentation in the revised manuscript.

Figure 3: Can you really say that these are PDFs? You are showing number of rainfall events in different bins? Or at least better visual presentation should be made since it is not easy to see which dataset yields better agreement with observed data. Additionally, can you add a summary of these differences between models and observed data? Thus, which model/dataset yields the best fit to the observed data.

We agree that the results in Figure 3 are not PDFs of the daily precipitation amounts, but histograms of the frequency of precipitation in different ranges which can reflect the PDFs. We will revise our wording and Figure 3 to improve the readability of the results. NMIC yields the best fit to the observed data, and it is reasonable to believe that NMIC is the closest to the real situation of areal average precipitation field among the four gridded products. We will add more clarification on this point and add a table including the results of Figure 2 and Figure 3 to make a summary.

Figure 4: I am sorry but I cannot understand this figure since obviously I do not understand correctly what should the ARF be according to your study. Perhaps this is related to the definition in L158-162 and Eq. 3 that should be improved.
What is usually defined as areal reduction factor can be seen here (for example):
https://www.sciencedirect.com/science/article/abs/pii/S0022169418301999.
Thus, something different that is shown in Figure 4. For example, the ARF should be a value between 0 and 1. I suggest that authors try to make this a bit easier to understand (what is shown here) for the reader.

We agree that the original definition of ARF is the ratio between the areal average rainfall and the point rainfall, to obtain areal rainfall from point rainfall values of specified durations and return periods. And generally, the ARF varies between 0 and 1. In the manuscript we defined ARF as the ratio of gridded precipitation metrics/rainfall erosivity over gauge-based precipitation metrics/rainfall erosivity, which can be confusing. We intend to change the index name to *ratio* in the revised version.

L289: Any specific reason for such behavior?

As mentioned in the discussion section, the difference between the western and eastern region mainly comes from the station density used for generating the gridded dataset, and the precipitation characteristics. The eastern region has a higher station density (> 4 stations per 10,000 square kilometers) and relatively humid climate (precipitation > 400 mm/y), whereas in the western region the density of stations is quite low and the climate is relative arid (< 1 station per 10,000 square kilometers, precipitation < 400 mm/y). The more stations were used, the more accurate the interpolation are, the better the scale discrepancy between gridded and gauge precipitation can be simulated, and the better the linear relationships are (Line 347-257). Large temporal and spatial variability of precipitation in arid western region also increases the relative error of interpolation, and the relationship between gridded gauge precipitation is more difficult

to describe. Besides, among four gridded products, different number of stations and interpolation methods were used (Table 1), which lead to the different performances of correction factors. In addition to large random errors affecting the effect of linear regression, some correction factors in western region are close to 1, so the correction do not improve the results. We will clarify this more clearly in the Discussion section in the revised version. There are some mistakes about the number of stations used for the generation of the gridded products, and we will correct them in the revised manuscript.

Figure 5: The resolution of this figure is quite low. I would suggest to add the number of points in all plots, since it seems that some cases (h) or (d) have relatively low number of points compared to the size of the investigated area. Or is this already written in the caption where "grids" is used? Moreover, grids or grid cells?

Since it was assumed that the grid cells containing meteorological stations can better represents the relationship between $R_{gri}$ and $R_{ref}$ than those without meteorological stations, only grid cells containing stations were used to establish and evaluate the correction factors. Due to the sparse stations in the western region, the number of points in (b), (d), (f) or (h) is much less than that in the cases of eastern region. Moreover, due to the different spatial resolutions of different gridded datasets, the number of non-repeated grid cells selected in each case is also different. We have written the numbers of used grid cells in the caption, and we will change "grids" to "grid cells".

Table 5: Correction in some cases leads to worse results? What is then the rationale behind adopting such "corrections" factors if the final result is even worse than without these factors.

Please see the responses to the general comment and the comment on L289.

Figure 7: I suggest to add a map that shows the difference between the erosivity map after applying correction and the Yue map.

We agree.

[Figure]

**Figure 7. Spatial distribution of the R-factor and 1-in-10-year event EI$_{30}$ using CN05.1 with bias correction factors (a and c), and the difference (b and d) in comparison with the original R-factor map (Yue et al., 2020b).**

In Figure 7, (b) and (d) show that the erosivity maps with bias correction factors using CN05.1 still have marked discrepancy in comparison the with the original R-factor map of Yue et al. (2020b). The difference remained because the same correction factor value was applied to all grid cells in the eastern and western regions. However, the accuracy of these maps generated with CN05.1 has been noticeably improved. Using the bias correction factors to reduce the bias is necessary only when gridded precipitation products are only data source for erosivity estimation.

Discussion: What not merging results and discussion since you already have some discussion in the results section? And then perhaps also the results section would be easier to read and understand.

This is a suggestion worth considering. We will carefully think how to better organize the structure of the manuscript.

L385: "Reductions": gauge data compared to grid data or grid data compared to gauge data?

Thank you for pointing this out. It should be "compared to gauge data, gridded products reduce the no-rain days and heavy precipitation days, and increase light precipitation days."

L400: What is the purpose of using a correction factor if it does not yield improved performance? Is there any alternative, a better method that should be elaborated?

Please see the response to the comments on Table 5.

Conclusions: What are the practical conclusions of this study that could be useful for people dealing with rainfall erosivity in other parts of the world? What is the main take home message?
Please see the responses to the general comment.

---

## Author Comment (AC3) · 5 Mar 2021

Reviewer's original comment in black
Our response to each comment in blue

This paper is well written with appropriate methods and datasets. However, the focus is on the comparison of different gridded rainfall datasets rather than the actual estimation (methods) on rainfall erosivity. Thus, I suggest to narrow down the title to truly reflect the actual content of the manuscript, the current title (Rainfall erosivity estimation using gridded daily precipitation datasets) is too broad. I also suggest the authors to clearly specify the differences of this research from previous studies and the implications to international readers, and add the aims and objectives in the last paragraph under the Introduction section.

Thank you for the review and valuable comments.

We will revise the objectives of this study in the revised version to make these clearer as we have found that the objectives were not clearly stated in the manuscript as follows:
(1) to contrast the gridded daily precipitation products with gauge data in terms of PDFs and extreme precipitation amounts, and to evaluate the smoothing effect of interpolation when areal precipitation for grid cells were generated using point (gauge) observations;
(2) to evaluate the magnitude of underestimation of rainfall erosivity calculated using gridded daily precipitation products compared with that produced by spatial interpolation of rainfall erosivity computed using point (gauge) observations;
(3) to establish bias correction factors to improve the accuracy of rainfall erosivity maps where only gridded precipitation products were available for estimating rainfall erosivity over large areas.

Comparing different gridded rainfall products, or identifying which gridded products is the best for China, is not one of our research objectives. We aim to explore the differences between various gridded data sources and how they differ from gauge data, and give the point that gridded data is inherently less extreme, which leads to underestimation of rainfall erosivity calculated using gridded products.

The difference between this research and previous studies and the implications of this research to a wider community internationally are:

(1) because the gridded precipitation products (represent areal average precipitation over a grid cell) are generally unable to capture the extreme precipitation amount, compared to gauge observations, care needs to be taken to avoid systematic bias when using gauge-based, satellite-based gridded precipitation products, reanalysis products and outputs of climate models as inputs to nonlinear models often found in hydrology and agriculture;

(2) when empirical models, such as the daily rainfall erosivity model considered in this study, are developed based on point observations, and gridded precipitation is the only available data source, bias correction is absolutely needed, or alternatively the model needs to be re-calibrated at a commensurate spatial scale.

We intend to change the title to: Evaluation of gridded precipitation products for rainfall erosivity estimation.

Some minor issues for authors consideration: Line 203: please differentiate the parameters in Eq. 9 (Rref = a Rref) Fig. 3-6: improve/change the fonts to make the figures clearer Check the unit for the R-factor: is hm-2 really km-2 or just use ha.

Sorry for our mistake. The equation 9 should be $R_{gri} = a \cdot R_{ref}$.
This typo will be corrected in the revised manuscript.